# Dissecting the Biological Functions of Various Isoforms of Ferredoxin Reductase for Cell Survival and DNA Damage Response

**DOI:** 10.3390/cells15010062

**Published:** 2025-12-29

**Authors:** Ken-ichi Nakajima, Shakur Mohibi, Kyle Hong, Xinbin Chen, Jin Zhang

**Affiliations:** Comparative Oncology Laboratory, Schools of Veterinary Medicine and Medicine, University of California, Davis, CA 95616, USA; kennakajima@ucdavis.edu (K.-i.N.); smohibi@antibodiesinc.com (S.M.); kkhong@ucdavis.edu (K.H.)

**Keywords:** ferredoxin reductase, FDXR isoforms, mitochondria, p53, DNA damage response

## Abstract

The ferredoxin reductase *(FDXR*) gene is expressed as seven isoforms: 1–6 by alternative splicing and 7 by an alternative promoter according to the Entrez Gene Database. Previous studies showed that FDXR, primarily the mitochondrial isoform 1, plays a role in biosynthesis of sterols, heme, and iron–sulfur clusters. However, the biological functions of FDXR isoforms 3–7 have not been characterized. Here, we first examined the expression profile of various FDXR isoforms. We found that isoform 1 is the most abundant one, accounting for ~70% of total FDXR, whereas isoforms 4 and 7 account for ~10% and ~7%, respectively. We found that isoforms 1 and 4 are mainly localized in the mitochondria, whereas isoform 7, which lacks a mitochondria localization signal (MLS), is expressed in the cytosol. We also found that like the promoter 1 for isoforms 1-6, the P2 promoter for isoform 7 can be induced by DNA damage in a p53-dependent manner. To determine isoform-specific activity, we generated multiple MCF7 cell lines in which one or more FDXR isoforms are knocked out. While total *FDXR*-KO MCF7 cells are non-viable, cells deficient in isoforms 1–6, isoform 4, or isoform 7 remain viable but are defective in cell proliferation, DNA damage response, and repair. These data suggest that each FDXR isoform contributes to cell survival and that isoform 7 has extra-mitochondrial activity that may be sufficient for cell survival.

## 1. Introduction

Ferredoxin reductase (FDXR) [1], also called adrenodoxin reductase (ADXR), is a mitochondrial flavoprotein and the sole human ferredoxin reductase that transfers electrons from nicotinamide adenine dinucleotide phosphate (NADPH) to the two human ferredoxin proteins, FDX1 and FDX2 [2,3]. FDX1 transfers electrons to type I cytochrome P450 enzymes (a mitochondrial form) that are involved in the metabolism of steroid hormones, bile acids, and vitamins A and D [4,5,6]. FDX2 donates electrons to the Fe-S biogenesis machinery for the synthesis of mitochondrial iron–sulfur clusters (ISCs) [7,8]. As such, FDXR participates in these pathways and plays a crucial role in both steroid hormone synthesis and iron–sulfur cluster (ISC) biogenesis. Due to its critical role in the electron transport chain, congenital FDXR mutations lead to mitochondrial disorders in humans [9,10], a group of genetic conditions that disrupt the body’s ability to produce energy [11]. The symptoms from patients with a FDXR germline mutation include optic atrophy, visual dysfunction, hearing loss, movement disorder, and global developmental delay [9,12,13]. In addition, a recent study found that FDXR mutation leads to adrenal insufficiency and atypical sexual development in patients [14], which is closely related to the steroid hormone synthesis. Moreover, we and others found that FDXR deficiency leads to abnormal iron accumulation in mitochondria and altered lipid metabolism [10,15,16]. Notably, the clinical characteristics of FDXR germline mutations are likely due to impaired electron transport to FDX1 or FDX2. Indeed, patients with FDX2 germline mutations exhibit similar mitochondrial disorders [17,18,19]. Additionally, a meeting report indicates that FDX1 germline mutations lead to an atypical form of congenital adrenal hyperplasia in humans (Clément Janot, ESPE Abstracts 98HA1).

Although it is known that FDXR plays a critical role in maintaining mitochondrial function, its comprehensive physiological role is still not fully understood. Interestingly, FDXR transcripts can be detected in the blood of patients receiving radiotherapy in a dose-dependent manner and, thus, are considered one of the most accurate markers for estimating radiation doses in radiotherapy [20,21,22,23]. In addition, we and others found that the FDXR transcript is up-regulated by p53 in response to DNA damage and stress signals and mediates p53-dependent tumor suppression through FDX2 and IRP2 [15,24,25]. Moreover, FDXR was also found to interact with the Fhit protein [26], a tumor suppressor, to modulate apoptosis, potentially through the regulation of ROS levels. Furthermore, we previously found that mice deficient in *Fdxr* die on embryonic days 7.0–8.0 [15]. Similarly, *FDXR*-KO cells cannot be generated by CRISPR-Cas9 [15]. In line with this, a previous study showed that the null mutation of the dare gene, a homolog of human FDXR, undergoes developmental arrest during the second larval instar or at the second larval molt, and intermediate severity mutations of the dare gene are delayed in pupariation, which can be rescued by feeding mutant larvae the insect steroid hormone [27]. These observations suggest that FDXR plays an important role in stress response, embryonic development, and efficient cell growth/survival. However, the underlying mechanism by which FDXR mediates all these biological functions has not been fully characterized.

To better understand the molecular basis of FDXR, we sought to characterize its various isoforms along with their expression profiles and potential biological functions in cells. According to the Entrez Gene database, the human *FDXR* gene (gene ID: 2232), located on chromosome 17q25.1 [28], is found to be expressed as eight isoforms due to alternative splicing and alternative use of two promoters. Isoforms 1–7 produce proteins, whereas isoform 8 (NR_047576.3) is a non-coding RNA transcript and does not encode a protein. While most studies have focused on the role of isoforms 1 and 2 in synthesis of steroids, sterol, and Fe/S clusters [29,30,31], the biological function of the remaining FDXR isoforms remains to be characterized. Interestingly, a recent study showed that all FDXR isoforms are induced in response to radiation exposure to varying extents; however, the biological significance of this induction remains unknown [23]. In the present study, we sought to examine the expression pattern and biological function of various *FDXR* isoforms, with a focus on three highly expressed protein-coding isoforms 1, 4, and 7. We showed that isoforms 1 and 4 are mainly localized in the mitochondria, whereas isoform 7 is localized in the cytosol. We also showed that although total *FDXR*-KO cells are non-viable, isoform-specific *FDXR*-KO MCF7 cells that are deficient in isoforms 1–6, or isoform 4 or isoform 7, remain viable. Furthermore, we showed that cells deficient in one or more FDXR isoforms are deficient in DNA damage response and prone to cell death. These data suggest that one FDXR isoform is sufficient for cell survival, and FDXR may have activity outside of mitochondria.

## 2. Materials and Methods

### 2.1. Reagents

Dulbecco’s modified minimal essential medium (DMEM), DreamTaq DNA polymerase, RevertAid reverse transcriptase, Trizol LS reagent, ProLong Gold with DAPI, MitoTracker deep red, Lipofectamine RNAiMAX, fetal bovine serum (FBS), and the Mitochondria Isolation Kit for Cultured Cells were purchased from Thermo Fisher Scientific (Waltham, MA, USA). Penicillin and streptomycin were purchased from RPI Research Products (Mt. Prospect, IL, USA) and Gemini Bio-products (West Sacramento, CA, USA), respectively. Trypsin was purchased from VWR (Radnor, PA, USA). Real-time quantitative PCR master mix was purchased from Applied Biosystems (Waltham, MA, USA). JetPRIME transfection reagent was purchased from Polyplus (Illkirch, France). Rabbit polyclonal anti-FDXR antibody (catalog number: 15584-1-AP) and rabbit polyclonal anti-Flag antibody (catalog number: 20543-1-AP) were purchased from ProteinTech (Rosemont, IL, USA). Mouse monoclonal anti-Flag M2 antibody (catalog number: F1804) was purchased from Sigma-Aldrich (St. Louis, MO, USA). Mouse monoclonal anti-actin antibody (catalog number: sc-47778) and mouse monoclonal anti-FDXR antibody (catalog number: sc-374436), p21 (catalog number: sc-53870), and cyclin A (catalog number: sc-271682) were purchased from Santa Cruz Biotechnology (Santa Cruz, CA, USA). Rabbit monoclonal anti-GAPDH antibody (catalog number: 2118), rabbit monoclonal anti-SDHA antibody (catalog number: 11998), p-CHK2 (threonine 68) (catalog number: 2661), p-CHK2 (serine 19) (catalog number: 2666), and total CHK2 (catalog number: 2662) were purchased from Cell Signaling Technology (Danvers, MA, USA). CellTiter-Glo 3D cell viability assay reagent was purchased from Promega (Madison WI, USA). The protease inhibitor cocktail was purchased from MedChemExpress (Monmouth Junction, NJ, USA). Custom siRNA duplexes were purchased from Horizon Discovery Ltd. (Cambridge, UK). All other chemicals were purchased from Millipore-Sigma (Burlington, MA, USA) or Fisher Scientific unless otherwise specified.

### 2.2. Cell Culture and Cell Line Generation

Cells from this study were originally obtained from the ATCC and used below passage 25. MCF7 (ATCC HTB-22); HCT 116 (ATCC CCL-247); Hep G2 (ATCC HB-8065); 4T1 (ATCC CRL-2539). MCF7, HCT116, HepG2, and 4T1 cells and their derivatives were cultured in DMEM (Dulbecco’s modified Eagle’s medium, Life Technologies, Carlsbad, CA, USA) supplemented with 10% fetal bovine serum (Life Technologies). *p53*-KO MCF7 and HCT116 cells were previously generated [15,32].

To generate isoform-specific *FDXR* KO cell lines, CRISPR-Cas9 technology was used. Briefly, MCF7 cells were transfected with a pSpCas9(BB)–2A-Puro vector expressing a guide RNA and then selected with puromycin for 2–3 wks. An individual clone was picked up and confirmed by genotyping and sequence analysis. The gRNA and genotyping primer sequences are listed in Table 1 and Table 2.

To generate a stable 4T1 cell line in which the endogenous *Fdxr* gene was ablated but expressed an HA-tagged human protein, 4T1 cells were transfected with two pSpCas9(BB)-2A-Puro vectors that expressed guide RNAs targeting the mouse *Fdxr* gene together with a pcDNA4 vector expressing HA-tagged human FDXR. Cells were then selected with puromycin and zeocin for 2–3 wks. A mixture of resistant cells was collected and saved. The expression of endogenous mouse FDXR and exogenous HA-tagged human FDXR was confirmed by Western blotting. The gRNA sequences that target the mouse *Fdxr* gene are listed in Table 3.

### 2.3. RNA Interference (RNAi)

MCF7 (both wild-type and exon/isoform KO) cells were transfected with 25 nM siRNA against the indicated FDXR isoform using Lipofectamine RNAiMAX, according to the manufacturer’s protocol. The siRNA sequences are listed in Table 4.

### 2.4. Immunofluorescence Microscopy and Phase-Contrast Microscopy

MCF7 cells were transfected with 3xFlag-tagged FDXR isoform 1, isoform 4, or isoform 7 using jetPRIME transfection reagent, according to the manufacturer’s protocol. At 24 h after transfection, the cells were incubated with 200 nM MitoTracker deep red for 30 min. The cells were fixed with 3.7% formaldehyde in PBS and permeabilized with 0.2% Triton X-100 in PBS and then stained with mouse anti-Flag antibody (1:200), followed by FITC-conjugated anti-mouse IgG antibody. The cells were mounted with ProLong gold with DAPI and observed using a Leica SP8 confocal microscope with a 40×or 63× oil immersion lens. To take phase-contrast microscopic images, an inverted microscope equipped with a Nikon Eclipse TS100 camera was used.

### 2.5. Western Blotting Analysis

Western blot analysis was performed as previously described [33]. Briefly, cells were washed with phosphate-buffered saline (PBS), lysed with 2× SDS lysis buffer, and boiled for 6 min at 95 °C. The cell lysates were then subjected to SDS-PAGE and subsequently transferred onto a nitrocellulose membrane using a wet transfer apparatus (Idea Scientific, Minneapolis, MN, USA). The membranes were incubated with primary and secondary antibodies, followed by detection with enhanced chemiluminescence and visualized by the VisionWorksLS software (Analytik Jena US, Upland, CA, USA)

### 2.6. Reverse Transcription–Polymerase Chain Reaction (RT-PCR) and Quantitative PCR (qRT-PCR)

Total RNA was extracted from cells using Trizol LS reagent according to the manufacturer’s protocol. First-strand cDNA was synthesized with RevertAid reverse transcriptase according to the manufacturer’s protocol. Standard RT-PCR was performed with the DreamTaq reagent with isoform-specific primer pairs. Real-time quantitative PCR was performed with 20 μL reactions using 2X QPCR SYBR Green Mix (Applied Biosystems, Waltham, MA, USA) with 5 μM primers. The reactions were run on a StepOne Plus (Applied Biosystems) using a two-step cycling program: 95 °C for 15 min, followed by 40 cycles at 95 °C for 15 s, 60 °C for 30 s, and 68 °C for 30 s. A melt curve (57–95 °C) was generated at the end of each run to verify the specificity. The primer sequences for qRT-PCR and RT-PCR are listed in Table 5 and Table 6.

### 2.7. Chromatin Immunoprecipitation (ChIP)

A ChIP assay was performed as described previously [34]. Briefly, chromatin was cross-linked by incubating the cells with 1% formaldehyde in PBS. The cells were then lysed, and the lysates were sonicated to yield 200–1000 bp DNA fragments and immunoprecipitated with a control IgG or a monoclonal antibody against p53. After reverse cross-linking and phenol/chloroform extraction, the DNA fragments were purified, followed by PCR to assess the enriched DNA fragments. p21 and GAPDH were used as positive and negative controls, respectively. The primer sequences used for the ChIP assays are listed in Table 7.

### 2.8. Colony Formation Assay

Five hundred or fifteen hundred cells were seeded into a 6-well plate and cultured for 14–17 days. The cell colonies were fixed with methanol/glacial acetic acid (7:1) and stained with a 0.1% crystal violet solution.

### 2.9. Recombinant Protein Expression

*Escherichia coli* BL21 cells were transformed with the pGEX-4T-1 vector expressing FLAG-tagged FDXR isoform 1 or isoform 7. The transformants were grown in LB medium until the OD600 reached approximately 0.5, followed by protein induction with 0.2 mM IPTG at 25 °C overnight. The bacterial cells were harvested, lysed using SDS-PAGE lysis buffer, and subjected to SDS-PAGE and Western blotting, as described elsewhere.

### 2.10. Cell Viability Assay

Cells were seeded into a 96-well plate at a density of 4000 cells/well with 100 µL of DMEM. An amount of 50 µL of the culture medium was removed, and then 50 µL of CellTiter Glo 3D reagent was added. Luminescence was measured using a SpectraMax Gemini XS fluorescence/luminescence microplate reader (Molecular Devices, San Jose, CA, USA). The luminescence in the control cells was set to 100%, and the relative cell survival was calculated as a ratio compared with the control.

### 2.11. Statistical Analysis

The results are shown as means ± SEM. Two-tailed Student’s *t*-test was used for the statistical analyses. A *p*-value of less than 0.05 was considered statistically significant.

## 3. Results

### 3.1. Expression Profiles of Various FDXR Isoforms

According to the Entrez Gene database, the *FDXR* gene (gene ID: 2232) contains 12 exons and is expressed as eight isoforms. Among them, isoforms 1–7 produce proteins, whereas isoform 8 (NR_047576.3) is a non-coding RNA transcript. Since isoform 8 does not produce a protein, it is not examined in the current study. Isoforms 1–6 were generated through alternative splicing, whereas isoform 7 was produced through an intron promoter (Figure 1A). Isoform 1 is considered the reference FDXR (Figure 1B). When compared with isoform 1, isoform 4 has an insertion and substitution right after its mitochondrial localization signal (MLS), whereas isoform 7 has a complete loss of MLS (Appendix A); isoform 2 has an insertion in the NADPH domain; isoform 3 has an insertion in the first flavin adenine dinucleotide (FAD)-binding domain; and isoforms 5–6 have a deletion in their first FAD-binding domain (Figure 1B). The sequence alignment for FDXR isoforms 1–7 is shown in Supplemental Figure 1. We would like to note that the sequences of isoforms 4 and 7 in the Entrez Gene database were reversed when compared with the ones in Uniprot. However, the sequences of isoforms 4 and 7 in this manuscript were based on those from the Entrez Gene database.

To determine the expression profiles of the seven FDXR transcripts by qRT-PCR in various cells, three cell lines were used: colorectal cancer cell line HCT116; breast cancer cell line MCF7; and liver cancer cell line HepG2. We found that in all three cell lines, isoform 1 is the most abundant isoform, accounting for ~70% of the total isoforms, followed by isoforms 4 and 7, which account for approximately 10% and 7%, respectively (Figure 1C–E). The remaining 10% is composed of isoforms 2–3 and 5–6, ranging from ~1% to 3% (Figure 1C–E). Thus, isoforms 1, 4, and 7 are the predominant isoforms expressed in tumor cells. We would like to note that, based on the Uniprot database, the predicted MLS of the FDXR protein is from aa 1 to 32. When compared with isoform 1, isoform 4 contains a 31-amino acid insertion and a 6-amino acid substitution between residues 29 and 34 (Figure 1). In addition, isoform 7 lacks the first 59 amino acids, including MLS, but includes an additional 7-amino acid extension at its N-terminus (Appendix A). Thus, it is possible that both isoforms 4 and 7 have an alteration in MLS. To test this, we generated a vector expressing C-terminal Flag-tagged isoform 1, 4, or 7, which was then transfected into 293T cells. We found that when detected by Flag or FDXR antibody, isoforms 1, 4, and 7 were each expressed as a major polypeptide band with different electrophoretic mobility: Isoforms 1 and 7 migrated similarly, whereas isoform 4 migrated more slowly, likely due to the insertion at its N-terminus (Figure 1F). We also observed that isoforms 1 and 4 were expressed as two polypeptides: one major polypeptide and a second one of much lower intensity, likely due to the cleavage of MLS. In contrast, isoform 7 was expressed as a single polypeptide and may not undergo MLS cleavage. To determine whether isoform 7 migrated similarly to the cleaved isoform 1, we expressed isoform 1 and 7 in *Escherichia coli* BL21, which do not have mitochondrial import machinery or proteases to process MLS. As expected, we found that the uncleaved isoform 1 expressed in BL21 migrated more slowly than isoform 7 when detected by either Flag or FDXR antibody (Figure 1G). These data suggest that the isoform 1 protein detected in 293T cells represented the cleaved form, which migrated similarly to isoform 7.

Next, an immunofluorescence assay was performed to visualize the localization of isoforms 1, 4, and 7. We found that isoform 1 colocalized with MitoTracker, suggesting that isoform 1 is mainly localized in mitochondria (Figure 1H, isoform 1 panel). Interestingly, isoform 4 also colocalized with MitoTracker, indicating that isoform 4 is also mainly localized in mitochondria (Figure 1H, isoform 4 panel). We postulate that due to the presence of MLS, isoform 4, like isoform 1, can enter the mitochondria. However, the MLS cleavage site in isoform 4 was impaired due to sequence insertion and substitution (Appendix A), which resulted in the uncleaved isoform 4 in mitochondria. This may explain why the major isoform 4 peptide migrates more slowly than isoforms 1 and 7 (Figure 1F). Furthermore, isoform 7, which does not contain MLS, did not colocalize with MitoTracker and was mainly localized in the cytosol (Figure 1H, isoform 7 panel).

### 3.2. All Seven FDXR Isoforms Are Induced by DNA Damage in a p53-Dependent Manner

We and others have shown previously that FDXR, that is, the reference isoform 1, is up-regulated by tumor suppressor p53 in response to DNA damage and mediates p53-dependent tumor suppression [24,25]. Thus, we determined whether other FDXR isoforms are regulated by DNA damage. To this end, HCT116, MCF7, and HepG2 cells were mock-treated or treated with doxorubicin (DOX) or camptothecin (CPT), followed by qRT-PCR, to measure the levels of each FDXR isoform. Doxorubicin is a topoisomerase II inhibitor, whereas camptothecin is a topoisomerase I inhibitor [35], both of which are DNA-damaging agents and known to activate p53 [36,37]. As shown in Figure 2A–C, DNA damage drastically increased the transcripts of all FDXR isoforms in all three cell lines, suggesting that all FDXR isoforms are up-regulated by DNA damage. Notably, isoform 7, which is driven by the P2 promoter located in intron 2, was also induced by camptothecin or doxorubicin in all three cell lines (Figure 2A–C). Although the P1 promoter of the *FDXR* gene is known to contain a p53 response element [24,25], it is uncertain whether p53 transactivates isoform 7 via the P2 promoter. To test this, isogenic control and *p53*-KO MCF7 and HCT116 cells were used to measure the induction of isoforms 1, 4, and 7 upon treatment with doxorubicin. We found that in response to doxorubicin treatment, the transcript level of total FDXR, isoform 1, 4, or 7 was markedly increased in both MCF7 and HCT116 cells (Figure 2D; compare lanes 1 and 5 with 2 and 6, respectively). In contrast, this up-regulation was abrogated in p53-KO MCF7 and HCT116 cells (Figure 2D; compare lanes 3 and 7 with 4 and 8, respectively), suggesting that p53 is required for induction of isoforms 1, 4, and 7 by DNA damage. Next, we searched for the promoter of isoform 7, which is located in intron 2, and found two potential p53 response elements (Figure 2E). Thus, a ChIP assay was performed with MCF7 cells mock-treated or treated with camptothecin. We found that p53 was able to bind to both putative p53 response elements on the FDXR P2 promoter (Figure 2F, p53RE#1 and p53RE#2 panels). The binding of p53 to the p21 promoter was used as a positive control (Figure 2F, p21 panel). In contrast, p53 was unable to bind to the GAPDH promoter (Figure 2F, GAPDH panel).

### 3.3. Generation and Validation of Isoform-Specific FDXR Knockout Cell Lines

Previously, we showed that loss of *Fdxr* leads to embryonic lethality, and no viable *Fdxr*-null mice were obtained [15]. Similarly, by using CRISPR-Cas9 technology, we were able to generate *FDXR^+/−^* but not *FDXR*-KO cells [15]. Since these knockout strategies ablated all FDXR isoforms, we sought to determine whether each isoform(s) is required for cell survival. To this end, MCF7 cells were used to generate isoform-specific knockout cell lines by using CRISPR-Cas9 to delete exon 1, 1′, or 2′ of the *FDXR* gene, respectively (Figure 3A–C). Consequently, the cell line with exon 1 deletion was expected to eliminate isoforms 1–6 and was named Ex1-KO MCF7 cells (Figure 1A and Figure 3A). The cell line with deletion of exon 1′ was expected to eliminate isoform 4 and was named Iso4-KO MCF7 cells (Figure 1A and Figure 3B). The cell line with deletion of exon 2′ was expected to eliminate isoform 7 and was named Iso7-KO MCF7 cells (Figure 1A and Figure 3C).

For each isoform-specific KO cell line, multiple clones were generated, and two clones from each genotype were selected to verify the deletion of exon 1, 1′, or 2′ by genotyping (Figure 3D). Surprisingly, isoform-specific KO MCF7 cell lines were successfully generated. Ex1-KO cells showed a deletion in exon 1, Iso4-KO cells had a deletion in exon 1′, and Iso7-KO cells had a deletion in exon 2′ (Figure 3E). DNA sequencing showed that both Ex1-KO cell lines (clones #25 and #26) had a 327 bp deletion spanning exon 1 and its surrounding region (Appendix A). For Iso4-KO MCF7 cells, clone #9 had a 21 bp deletion in exon 1′ and a 21 bp deletion in intron 1, whereas clone #12 had a 28bp deletion in exon 1′ (Supplemental Appendix A). For Iso7-KO cells, both clones had a 373 bp deletion spanning exon 2′ and its surrounding region (Supplemental Appendix A).

To measure the transcript levels of isoforms 1, 4, and 7 in each isoform-specific-KO cell, compared with isogenic control cells, RT-PCR was performed. We showed that in Ex1-KO cells, isoform 7 was detectable, whereas isoforms 1 and 4 were undetectable (Figure 3F; compare lanes 1–2 with 3–4). In Iso4-KO cells, isoforms 1 and 7 were detectable, whereas isoform 4 was undetectable (Figure 3F; compare lanes 1–2 with lanes 5–6). In Iso7-KO cells, isoforms 1 and 4 were detectable, whereas isoform 7 was undetectable (Figure 3F, compare lanes 1–2 with lanes 7–8). Next, the level of FDXR protein in various isoform-specific KO cells was measured and compared with that in isogenic control cells. We showed that the amount of FDXR protein in Ex1-KO cells was markedly decreased compared with isogenic control cells (Figure 3G; compare lanes 1–2 with lanes 3–4). This was due to the absence of isoforms 1 and 4 (Figure 3F), which is consistent with >80% of total FDXR (Figure 1C–E). Additionally, we showed that the amount of FDXR protein in Iso4- and Iso7-KO MCF7 cells were slightly reduced compared with the isogenic control cells (Figure 3G; compare lanes 1–2 with 5–8), which is also consistent with the observations that isoforms 4 and 7 only represent ~10% and 7% of total FDXR protein (Figure 1C–E), respectively.

### 3.4. Each FDXR Isoform Contributes to Cell Viability

To evaluate the role of each FDXR isoform in cell survival, a cell viability assay was performed with isogenic control cells and Ex1-KO, Iso4-KO, and Iso7-KO MCF7 cells. We found that cells deficient in isoforms 1–6 (Ex1-KO), isoform 4 (Iso4-KO), or isoform 7 (Iso7-KO) displayed reduced cell survival compared with isogenic control cells (Figure 4A). We also found that Ex1-KO cell survival was slightly poorer than that for Iso4- and Iso7-KO FDXR cells (Figure 4A).

Since we were able to generate Ex1-KO (lacking isoforms 1–6) in this study but not total *FDXR*-KO cells [15], we speculate that isoform 7 alone is sufficient for cell survival. To further test this, two independent siRNAs, which can target all FDXR isoforms, were designed and transfected into isogenic control, Ex1-KO, Iso4-KO, and Iso7-KO MCF7 cells, followed by measurement of cell viability and colony formation assays. As expected, FDXR siRNAs decreased the level of FDXR protein in isogenic control cells and further decreased the level of FXDR protein in isoform-specific KO (Ex1-KO, Iso4-KO, and Iso7-KO) cells (Figure 4B). Similarly, we found that knockdown of FDXR decreased the cell survival of isogenic control cells and further decreased the cell survival of isoform-specific (Ex1-KO, Iso4-KO, and Iso7-KO) cells (Figure 4C). Additionally, we found that knockdown of FDXR reduced the number of colonies of isogenic control cells and further decreased the number of colonies of isoform-specific KO (Ex1-KO, Iso4-KO, and Iso7-KO) cells (Figure 4D).

The above data suggest that the remaining FDXR isoform in each isoform-specific KO cell plays a critical role in maintaining cell proliferation. To test this, an siRNA that specifically targets isoform 7 was designed and then used for cell proliferation assays. We showed that knockdown of isoform 7 decreased the level of the FDXR protein in Ex1-KO and slightly in isogenic control cells (Figure 4E). We also showed that knockdown of isoform 7 markedly inhibited the cell proliferation of Ex1-KO cells (Figure 4F,G). However, we also noticed that the reduction in cell proliferation was much less pronounced in isogenic control cells (Figure 4F,G). These data are consistent with the observation that isoform 7 is the predominant isoform expressed in Ex1-KO cells, whereas it is only a minor isoform in isogenic control cells (Figure 3G). We also designed an siRNA, namely, si-Ex2, which targets isoforms 1 and 4, which was then transfected into isogenic control and Iso7-KO cells. As expected, si-Ex2 efficiently reduced the level of FDXR protein in both isogenic control and Iso7-KO cells (Figure 4H). Indeed, we found that knockdown of isoforms 1 and 4 greatly reduced the cell proliferation of both isogenic control and Iso7-KO cells (Figure 4I,J), consistent with the observation that isoforms 1 and 4 represent the majority of FDXR in both isogenic control and Iso7-KO cells (Figure 4H). Together, these data suggest that each FDXR isoform contributes to the overall cell survival.

### 3.5. Cells Deficient in One or More FDXR Isoforms Are Prone to DNA Damage-Induced Growth Suppression Along with Impaired DNA Damage Response

To determine the roles of various FDXR isoforms in DNA damage-induced growth suppression, a colony formation assay was performed with isogenic control cells and Ex1-KO, Iso4-KO, and Iso7-KO MCF7 cells mock-treated or treated with doxorubicin. We found that under non-stress conditions, the number of colonies was reduced most by Ex1-KO, followed by Iso7-KO and Iso4-KO, compared with isogenic control cells (Figure 5A), which is consistent with findings from short-term viability assays (Figure 4A). Upon treatment with doxorubicin, the number of colonies was markedly reduced by Ex1-KO, Iso4-KO, and Iso7-KO compared with isogenic control cells (Figure 5A).

As a reductase, loss of FDXR is known to enhance oxidative stress in the nucleus [25]. Thus, to examine whether loss of a FDXR isoform has an effect on DNA damage response, the level of γH2AX, a biomarker of DNA damage and repair [38], was measured in isogenic control and various isoform-specific KO cells mock-treated or treated with doxorubicin. We showed that upon treatment with doxorubicin, the level of FDXR protein was induced in Ex1-KO, Iso4-KO, and Iso7-KO cells, similar to isogenic control cells (Figure 5B), which is consistent with the observation that the level of FDXR transcripts was induced by DNA damage (Figure 2A–D). Surprisingly, we found that the level of γH2AX was much lower in Ex1-KO, Iso4-KO, and Iso7-KO cells than that in isogenic control cells, regardless of DNA damage treatment (Figure 5B), suggesting that DNA damage response may be impaired in Ex1-KO, Iso4-KO, and Iso7-KO cells. To verify this, we also examined phosphorylation of CHK2 at threonine 68 and serine 19 [39], which promote CHK2 dimerization and activation, thereby enabling cell cycle arrest to facilitate DNA repair. Interestingly, we found that phosphorylation of CHK2 at threonine 68 and serine 19 was markedly reduced in Iso4-KO and Iso7-KO cells, but moderately reduced in Ex1-KO cells, compared with isogenic control cells (Figure 5C). Since decreased CHK2 phosphorylation is associated with impaired cell cycle arrest, we examined the expression of p21 and cyclin A. We found that p21 levels, but not cyclin A levels, were decreased in Ex1-KO, Iso4-KO, and Iso7-KO cells compared with isogenic control cells (Figure 5D). To further evaluate the role of FDXR in DNA damage repair and apoptosis, we conducted a literature review and established two gene signatures: a nine-gene DNA repair signature (MGMT, APEX1, GADD45G, NUDT1, PCNA, FEN1, H2AFX, and WEE1) [21,40,41,42,43] and an eight-gene apoptosis signature (AIFM1, AKT1, APAF1, ATM, BAD, BAX, BCL2, and CASP3) [44,45,46,47]. We then examined the association of FDXR expression with these gene signatures across breast, colon, and liver cancers by performing Spearman correlation analysis using TCGA datasets. Interestingly, we found that FDXR was positively associated with the nine-gene signature involved in DNA repair and the eight-gene signature in breast, colon, and liver cancers (Appendix A–C). Similarly, FDXR was also found to be positively associated with the eight-gene signature involved in apoptosis in breast, colon, and liver cancers (Appendix A–F). Together, these findings suggest that loss of one or more FDXR isoforms may impair DNA damage response and repair, leading to increased cell death.

Previously, we showed that we were unable to generate a total *FDXR*-KO cell line or mouse [15]. Here, we showed that each FDXR isoform contributes to cell survival and DNA damage response (Figure 5A–D). These data prompted us to further verify the role of FDXR in cell survival, performing a rescue experiment. Specifically, we sought to ablate the endogenous mouse *Fdxr* gene in 4T1, a mouse mammary carcinoma cell line, and simultaneously express exogenous human FDXR protein. Briefly, 4T1 cells were co-transfected with two guide RNA plasmids targeting the mouse FDXR gene, along with a plasmid expressing HA-tagged human FDXR protein, followed by drug selection. After two weeks, a pool of viable cells was obtained, which was then tested for expression of endogenous and exogenous FDXR proteins. As shown in Figure 5E, we showed that these cells, named *FDXR*-KO+HA-FDXR 4T1 cells, did not express endogenous mouse FDXR, as confirmed by immunoblotting with an anti-FDXR antibody (Figure 5E). However, they do express HA-tagged human FDXR, which was detected using an anti-HA antibody (Figure 5E). Notably, the morphology of these cells was indistinguishable from isogenic control cells (Figure 5F). Next, we performed a cell viability assay by treating isogenic control and *FDXR*-KO+HA-FDXR 4T1 cells with or without camptothecin and doxorubicin. We found that cell viability was comparable between the isogenic control and *FDXR*-KO+HA-FDXR 4T1 cells (Figure 5G), indicating that the cell death mediated by loss of endogenous FDXR can be compensated for by exogenous FDXR protein. Together, these data further confirmed that at least one FDXR isoform is needed for maintaining cell survival.

## 4. Discussion

FDXR is known for its role as a reductase to reduce FDX1/2 and subsequently participates in the biogenesis of steroids, heme A, and ISC [2,3]. In addition, FDXR plays a critical role in iron homeostasis and contributes to p53-mediated tumor suppression [15]. However, it remains uncertain whether FDXR participates in other biological processes beyond those mentioned above. To better understand the biological function of FDXR, we examined the expression profile of seven FDXR isoforms. We found that isoform 1 is the predominant isoform and accounts for about 70% of total transcripts, followed by isoforms 4 and 7, ranging from 7 to 10% (Figure 1C–E). Sequence alignment indicated that compared with isoform 1, isoform 7 is produced by a P2 promoter and does not have an MLS, whereas isoform 4 contains an altered MLS with insertion and substitution (Appendix A). Immunofluorescence assay showed that isoform 7 is localized solely in the cytosol, while isoform 1 and isoform 4 are present in the mitochondria (Figure 1H). The finding that isoform 4 is present in mitochondria lets us speculate that the mitochondrial localization signal in isoform 4 can direct the protein to the mitochondria. However, since isoform 4 contains an insertion and two amino acid substitutions immediately upstream of the cleavage site, it may not be properly cleaved, which was also consistent with the observation that isoform 4 migrates more slowly than isoforms 1 and 7 (Figure 1F).

We reported previously that total *FDXR*-KO cells and mice are not viable [15], indicating that FDXR plays a critical role in development and cell survival. Surprisingly, in this study, we successfully generated various isoform-specific *FDXR*-KO MCF7 cell lines, including Ex1-KO cells, isoform 4-KO, and isoform 7-KO cells (Figure 3). As a result, Ex1-KO cells express only isoform 7, isoform 4-KO cells express isoforms 1 and 7, and isoform 7-KO cells express isoforms 1 and 4. To further explore the role of FDXR isoforms in cell viability, we designed specific siRNAs capable of targeting all FDXR isoforms or selectively targeting isoform 1/4 or isoform 7. We found that the cell viability of Ex1-KO, isoform 4-KO, and isoform 7-KO cells was suppressed by siRNAs targeting all FDXR isoforms (Figure 4B–D). Moreover, we showed that the cell viability of Ex1-KO cells was decreased by knockdown of isoform 7, whereas the cell viability of isoform 7-KO cells was decreased by knockdown of isoforms 1 and 4 (Figure 4E–J). Finally, we showed that ectopic expression of isoform 1 was able to rescue total *Fdxr*-KO in 4T1 cells (Figure 5E–G). These data indicate that each FDXR isoform contributes to cell survival and that isoform 7 alone is sufficient for cell survival.

In this study, we showed that Ex1-KO MCF7 cells remain viable, albeit proliferating more slowly than isogenic control, Iso1-KO, or Iso 7-KO MCF7 cells (Figure 4A). In addition, isoform-specific knockout cells have impaired DNA damage response (Figure 5B,C). Furthermore, exogenous FDXR can rescue cell death mediated by total FDXR knockout (Figure 5E–G). Ex1-KO MCF7 cells only express isoform 7, which lacks MLS and is expressed in the cytoplasm but not mitochondria (Figure 1H), suggesting that FDXR exerts activity outside mitochondria. Indeed, although the majority of FDXR protein localizes to the mitochondria, it is still possible that a portion of FDXR protein is retained in the cytoplasm and exerts non-mitochondrial functions. We postulated that the mitochondrial function of FDXR may not be essential for maintaining cell survival. Interestingly, isoforms 1, 4, and 7 contain the same NADP and FAD domains; these isoforms would exert the same reductase function. Thus, if a single FDXR isoform is sufficient to maintain cell survival, we postulate that the reductase activity, rather than the subcellular localization of the FDXR protein, is essential for cell survival. Patients with germline FDXR mutations are prone to mitochondrial disease, with some dying at a very young age. Additionally, FDXR mutations occur infrequently in the MLS, but primarily within the NADP and FAD domains, which are crucial for its enzymatic activity. We speculated that mutations essential for electron transfer likely disrupt FDXR’s reductase function, contributing to mitochondrial disease. Thus, further studies are warranted to identify FDXR binding partners in the cytoplasm, which are reduced by FDXR and contribute to cell survival.

In this study, we found that the level of γH2AX protein is lower in isoform-specific *FDXR*-KO cells compared with isogenic control cells under normal and DNA damage-induced conditions (Figure 5B). These data suggest that an adequate level of FDXR protein is required for proper DNA damage response and repair in cells. As a result, cells with low FDXR expression may fail to repair DNA properly and undergo apoptosis. Indeed, all isoform-specific *FDXR*-KO cells are prone to cell death induced by doxorubicin (Figure 5A). These findings are consistent with the observation that FDXR is one of the early genes to be up-regulated in response to radiation in humans [22,23]. Nevertheless, further studies are needed to address how FDXR participates in DNA damage response. It would also be interesting to determine whether the reductase activity is required for FDXR to participate in DNA damage response.

In summary, we have provided several novel insights into the roles of different FDXR isoforms in cell survival, DNA repair, and the cellular response to stress. Further studies are needed to elucidate how these isoforms contribute to both mitochondrial and non-mitochondrial functions in maintaining cellular homeostasis. A deeper understanding of these mechanisms may reveal new therapeutic opportunities for treating diseases associated with FDXR dysfunction, including various mitochondrial disorders and cancers.

## Figures and Tables

**Figure 1 cells-15-00062-f001:**
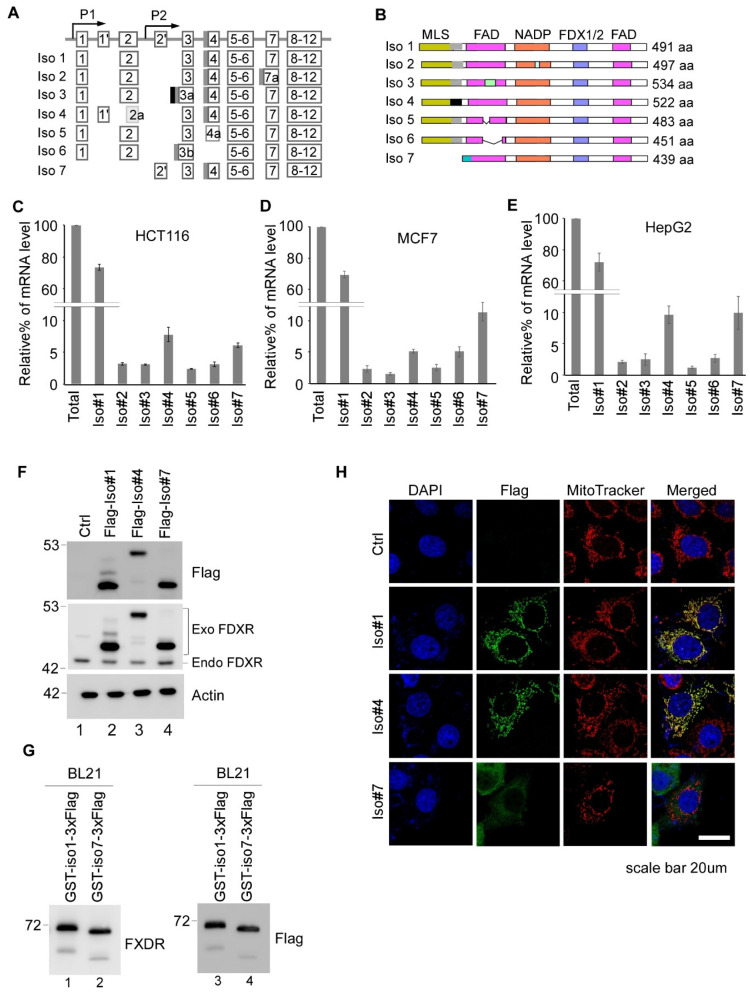
Expression profiles of various FDXR isoforms. (**A**) Schematic presentation of FDXR isoforms that are generated from alternative splicing or usage of different promoters. P1 and P2 promoters induce expression of isoforms 1–6 and isoform 7, respectively. The gene ID for each of the isoforms is NM_024417.5 for isoform 1; NM_004110.6 for isoform 2; NM_001258012.4 for isoform 3; NM_001258013.4 for isoform 4; NM_001258014.4 for isoform 5; NM_001258015.3 for isoform 6; and NM_001258016.3 for isoform 7. (**B**) The protein structures of various FDXR isoforms. MLS, mitochondria localization signal. FAD, FAD-binding domain. NADP, NAPDH-binding domain. FDX1/2, FDX1/2-binding domain. (**C**–**E**) qRT-PCR was performed in triplicate to measure the transcript levels of various FDXR isoforms in HCT116 (**C**), MCF7 (**D**), and HepG2 (**E**) cells. Total FDXR expression was used to normalize each isoform expression. (**F**) MCF7 cells were transiently transfected with an empty pcDNA3 vector or a vector expressing C-terminal Flag-tagged FDXR isoforms 1, 4, and 7 for 24 h, followed by Western blot analysis with antibodies against Flag, FDXR, or actin. (**G**) Recombinant FDXR isoform 1 or 7 produced in BL21 was subjected to Western blot analyses using FDXR (left panel) or Flag (right panel) antibody. (**H**) MCF7 cells transfected with a control vector or a vector expressing Flag-tagged isoforms 1, 4, and 7, followed by immunofluorescence using Flag antibody. Mitochondria and nuclei were visualized with MitoTracker and DAPI, respectively. Scale bar: 20 μm.

**Figure 2 cells-15-00062-f002:**
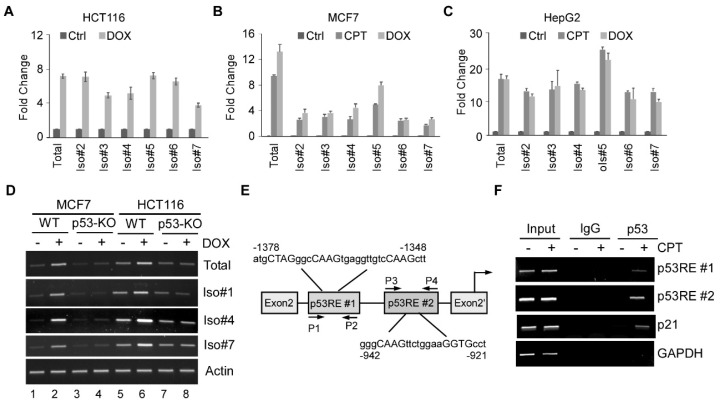
The transcripts of FDXR isoforms are up−regulated by DNA damage. (**A−C**) qRT-PCR was performed to measure the transcripts of various FDXR isoforms in HCT116 (**A**), MCF7 (**B**), and HepG2 (**C**) mock-treated or treated with doxorubicin or camptothecin. We used the total FDXR expression of the control condition (no treatment) to normalize the expression of each isoform in different drug treatments. (**D**) Isogenic control and *p53*−KO HCT116 and MCF7 cells were treated with or without doxorubicin, followed by RT−PCR to measure total FDXR, isoform 1, isoform 4, isoform 7, and actin transcripts. (**E**) Schematic presentation of predicted p53 response elements in the P2 promoter, as well as the locations of the primers for ChIP analyses. (**F**) MCF7 cells were mock-treated or treated with camptothecin, followed by ChIP analysis to measure the binding of endogenous p53 to the FDXR P2 promoter, p21 promoter, or GAPDH promoter.

**Figure 3 cells-15-00062-f003:**
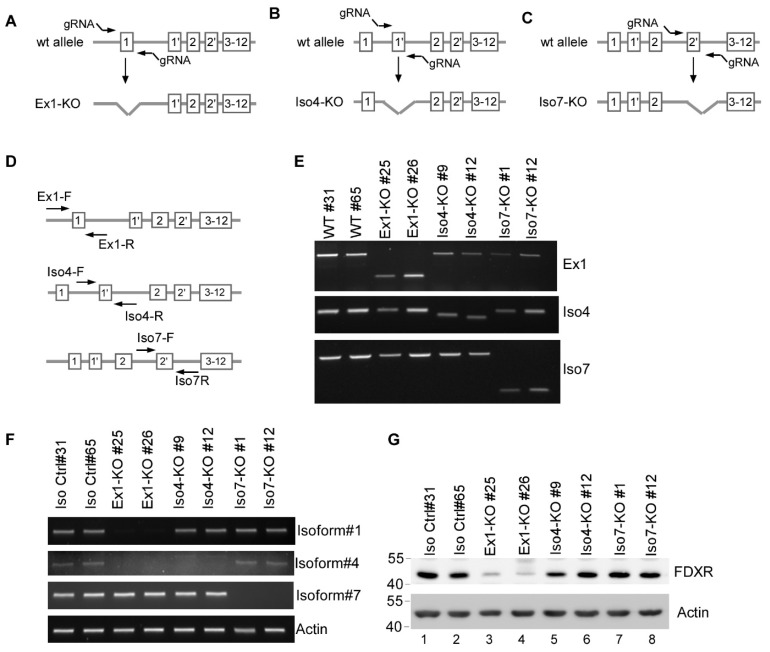
Generation and validation of isoform-specific FDXR knockout cell lines. (**A**–**C**) Strategies to generate isoform-specific FDXR knockout cell lines using CRISPR-Cas9 technology (**A**–**C**) and the gRNA location for Ex1-KO (**A**), isoform 4-KO (**B**), and isoform 7-KO (**C**). (**D**) A diagram showing the location of genotyping primers for Ex1-KO, isoform 4-KO, and isoform 7-KO. (**E**) PCR was performed to verify the specific deletions in Ex1-KO, isoform 4-KO, and isoform 7-KO cells. (**F**) The levels of total FDXR, isoform 1, isoform 4, isoform 7, and actin transcripts were measured in each of two isogenic controls, Ex1-KO, isoform 4-KO, and isoform 7-KO MCF7 cells. (**G**) The levels of FDXR and actin proteins were measured in isogenic control, Ex1-KO, isoform 4-KO, and isoform 7-KO MCF7 cells.

**Figure 4 cells-15-00062-f004:**
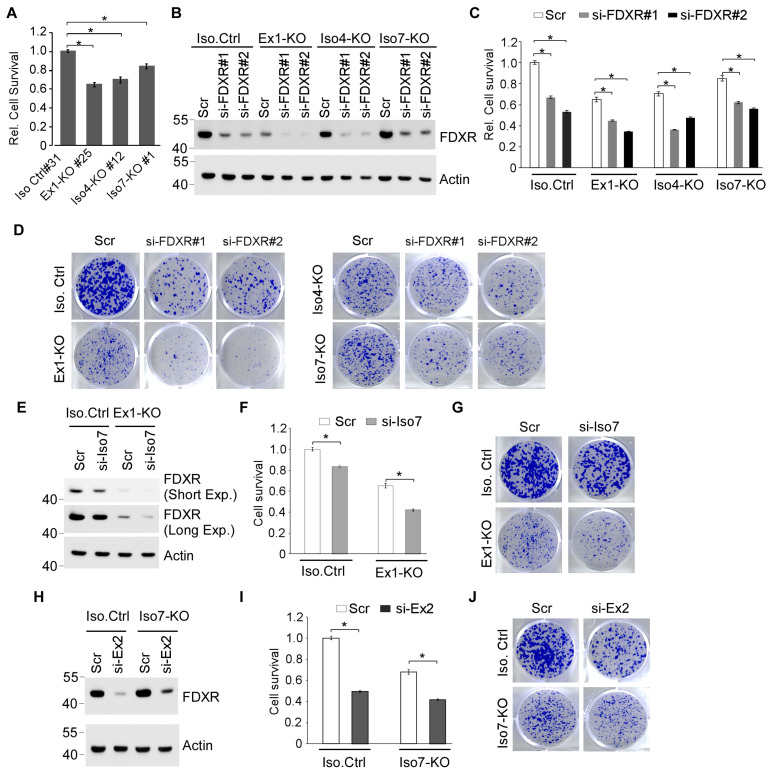
Each FDXR isoform contributes to cell viability. (**A**) Cell viability was measured in isogenic control and isoform-specific *FDXR*-KO MCF7 cells by performing CellTiter-Glo assays. The cell viability was arbitrarily set up as 1.0 in isogenic control cells, and relative cell survival was calculated as a ratio to control cells (*n* = 3). * indicates *p* < 0.05 by Student’s *t*-test. (**B**) The levels of FDXR and actin proteins were measured in isogenic control, Ex1-KO, isoform 4-KO, and isoform 7-KO MCF7 cells transfected with scrambled siRNA or siRNAs against total FDXR for 3 days. (**C**) CellTiter-Glo assay was performed in triplicate to measure cell viability using cells treated as in (**B**). * indicates *p* < 0.05 by Student’s *t*-test. (**D**) Colony formation assay was performed with the cells treated as in (**B**). (**E**) The levels of FDXR and actin proteins were measured in isogenic control and Ex1-KO MCF7 cells transfected with scrambled siRNA or siRNA against isoform 7 for 3 days. (**F**,**G**) CellTiter-Glo (**F**) and colony formation (**G**) assays were performed in triplicate using cells treated as in (**E**). * indicates *p* < 0.05 by Student’s *t*-test. (**H**) The levels of FDXR and actin proteins were measured in isogenic control and isoform 7-KO MCF7 cells transfected with scrambled siRNA or siRNA against isoforms 1 and 4 for 3 days. (**I**,**J**) CellTiter-Glo (**I**) and colony formation (**J**) assays were performed in triplicate using cells treated as in (**H**). * indicates *p* < 0.05 by Student’s *t*-test.

**Figure 5 cells-15-00062-f005:**
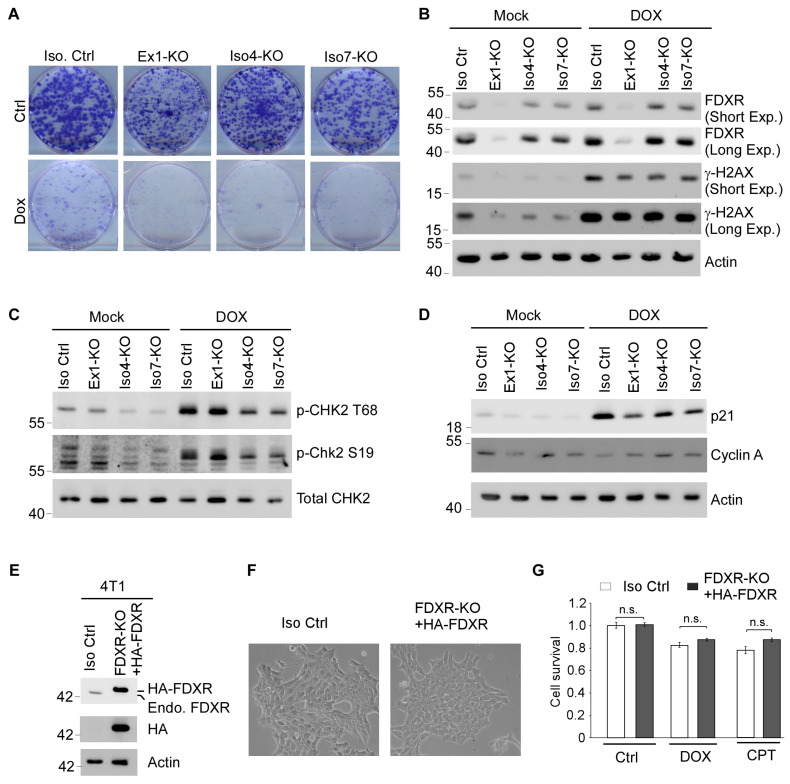
Cells deficient in one or more FDXR isoforms are prone to DNA damage-induced growth suppression along with impaired DNA damage response. (**A**) Colony formation assay was performed with isogenic control and isoform-specific *FDXR*-KO MCF7 cells treated with or without doxorubicin. (**B**) The levels of FDXR, g-H2Ax, and actin proteins were measured in isogenic control and isoform-specific *FDXR*-KO MCF7 cells treated with or without doxorubicin. (**C**) The level of phosphorylated CHK2 at threonine 68 and serine 19 and total CHK2 was examined by Western blot analysis in isogenic control and isoform-specific *FDXR*-KO MCF7 cells treated with or without doxorubicin. (**D**) The levels of p21, cyclin A, and actin were examined by Western blot analysis in isogenic control and isoform-specific *FDXR*-KO MCF7 cells treated with or without doxorubicin. (**E**) Isogenic control and *FDXR*-KO+HA-FDXR 4T1 cells were subjected to Western blot analyses using antibodies against HA, FDXR, and actin. (**F**) Phase-contrast microscopic images of isogenic control and *FDXR*-KO+HA-FDXR 4T1 cells. (**G**) Isogenic control and *FDXR*-KO+HA-FDXR 4T1 cells were mock-treated or treated with doxorubicin or camptothecin for 18 h, followed by CellTiter-Glo assay to measure cell viability (*n* = 3). n.s. indicates no significance by Student’s *t*-test.

**Table 1 cells-15-00062-t001:** The oligos used for sgRNA expression vectors.

Name	Sequence
Exon 1 sgRNA #1	Forward: 5′-caccgCGGGATTCTCTCGGGAGTCG-3′ Reverse: 5′-aaacCGACTCCCGAGAGAATCCCGc-3′
Exon 1 sgRNA #2	Forward: 5′-caccgCGAGATCCCGGTGGTGTAC-3′ Reverse: 5′-aaacGTACACCACCGGGATCTCGCc-3′
Isoform 4 sgRNA #1	Forward: 5′-caccgACTTCATCTGAACCCCCAA-3′ Reverse: 5′-aaacTTGGGGGTTCAGATGAAGTc-3′
Isoform 4 sgRNA #2	Forward: 5′-caccgTCAAGGCTTTGGCATTTGCA-3′ Reverse: 5′ -aaacTGCAAATGCCAAAGCCTTGAc-3′
Isoform 7 sgRNA #1	Forward: 5′-caccgGTTCAAACTGCTCGGCCTAG-3′ Reverse: 5′-aaacCTAGGCCGAGCAGTTTGAACc-3′
Isoform 7 sgRNA #2	Forward: 5′-caccgACATCCAAGGGTCTCAGGTT-3′ Reverse: 5′-aaacAACCTGAGACCCTTGGATGTc-3′

**Table 2 cells-15-00062-t002:** The primers used for genotyping cell lines.

Name	Sequence
Exon 1-KO genotyping	Forward: 5′-GCGTATACCCCGGATGCTC-3′ Reverse: 5′-TCACCTTTGTTGACCTCCGTC-3′
Isoform 4-KO genotyping	Forward: 5′-CCGTTGGAAGGATGTGGGAT-3′ Reverse: 5′-CCAGTCCCAAACCAACCTGA-3′
Isoform 7-KO genotyping	Forward: 5′-CGAGGGAACAGGAGCAGAAG-3′ Reverse: 5′-GCCTTGCTCAAAAATCTTCGC-3′

**Table 3 cells-15-00062-t003:** The oligos used for sgRNA expression vectors against mouse FDXR.

Name	Sequence
Exon 3 sgRNA #1	Forward: 5′-caccgCTTCTCGTAGATGTCTACGT-3′ Reverse: 5′-aaacACGTAGACATCTACGAGAAGc-3′
Exon 3 sgRNA #2	Forward: 5′-caccgCCAACACCACCCTCCGCACC-3′ Reverse: 5′-aaacGGTGCGGAGGGTGGTGTTGGc-3′

**Table 4 cells-15-00062-t004:** siRNA oligonucleotides.

Name	Sequence
Scrambled siRNA	5′-GCAGUGUCUCCACGCACUAdTdT-3′
siFDXR #1	5′-GCUCAGCAGCAUUGGGUAUUU-3′
siFDXR #2	5′-CACCAUUAAGGAGCUUCGG-3′
siIso7	5′-AGGAAAGAAUGGAAGAUAAUU-3′
siEx2	5′-CGGCCCAACACCUGCUAAAdTdT-3′

**Table 5 cells-15-00062-t005:** The primers used for real-time PCR.

Name	Sequence
FDXR (Total)	Forward: 5′-GTACAACGGGCTTCCTGAGA-3′ Reverse: 5′-CTCAGGTGGGGTCAGTAGGA-3′
FDXR (Iso#2)	Forward: 5′-CTGAGAACCAGGAGCTGGAG-3′ Reverse: 5′-GTCCGTTCTCTGGCACAAA-3′
FDXR (Iso#3)	Forward: 5′-GGTGGAAGCCTTGTGTTCT-3′ Reverse: 5′-GAGAGAGAGAGGCTGGGA-3′
FDXR (Iso#4)	Forward: 5′-TGAAGTAAGAGACCCTGCAAAT-3′ Reverse: 5′-ATATCCAACAGAAGCTGGAACT-3′
FDXR (Iso#5)	Forward: 5′-CTTCTACACGGCCCAACAC-3′ Reverse: 5′-AATGGGCCGTCTTCACCT-3′
FDXR (Iso#6)	Forward: 5′-GCCACCATTTCTCCACACAG-3′ Reverse: 5′-CCCCGTAGCTCTTCACCTC-3′
FDXR (Iso#7)	Forward: 5′-AGGTCAGCCACGAGAGATA-3′ Reverse: 5′-GCTCTCTGTCCTTATCTTCCATTC-3′

**Table 6 cells-15-00062-t006:** The primers used for RT-PCR.

Name	Sequence
FDXR (Total)	Forward: 5′-GTACAACGGGCTTCCTGAGA-3′ Reverse: 5′-CTCAGGTGGGGTCAGTAGGA-3′
FDXR (Iso#1)	Forward: 5′-GCAGTAGCTAGGAACAGATCC-3′ Reverse: 5′-TGTGTGGAGAAATGGTGGCAG-3′
FDXR (Iso#4)	Forward: 5′-GCTTCGCGCTGCTGGCGCTG-3′ Reverse: 5′-ATATCCAACAGAAGCTGGAACT-3′
FDXR (Iso#7)	Forward: 5′-AGGTCAGCCACGAGAGATA-3′ Reverse: 5′-GCTCTCTGTCCTTATCTTCCATTC-3′
Actin	Forward: 5′-CACTGTGCCCATCTACGAGG-3′ Reverse: 5′-TGGCCATCTCTTGCTCGAAG-3′

**Table 7 cells-15-00062-t007:** The primers used for ChIP.

Name	Sequence
p53RE #1	Forward: 5′-CCTCATGACAACCTGCAAAGC-3′ Reverse: 5′-TGTTCCCAGTAAAGCCTGCG-3′
p53RE #2	Forward: 5′-TGTCCCAGGCACAGAGAACT-3′ Reverse: 5′-TCTGATCGGGGAAGAGGAGG-3′
p21	Forward: 5′-CAGGCTGTGGCTCTGATTGG-3′ Reverse: 5′-TTCAGAGTAACAGGCTAAGG-3′
GAPDH	Forward: 5′-AAAAGCGGGGAGAAAGTAGG-3′ Reverse: 5′-AAGAAGATGCGGCTGACTGT-3′

## Data Availability

All study data are included within this article.

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
