# Peer review of "Dissecting the Biological Functions of Various Isoforms of Ferredoxin Reductase for Cell Survival and DNA Damage Response"

_cells, 2025, doi:10.3390/cells15010062_

Round 1

Reviewer 1 Report

Comments and Suggestions for Authors

This manuscript by Nakajima and colleagues looks at the roles of ferrodoxin isoforms in cell viability and growth and in the DNA damage response. The work seems to have been carefully carried out, on the whole, although I have a few criticisms which should be addressed. 

The argument that iso#7 is not located in the mitochondria is not convincing (Figure 1H) as the mito staining in the appropriate panel is totally different to other mito stained panels, unless the authors propose that isoform 7 disrupts mitochondria structure. In this panel isoform 7 looks the same as the mito staining. This needs to be clarified.

Simply using H2AX as a marker for DNA damage is not really sufficient. Staining for phosphorylation of other DNA repair proteins should be carried out. For example, blots should be carried out looking at the phosphorylation of either NBS1, RPA32, ATM or Chk1, Chk2 or KAP1.

I really don't see the point of the  experiment described in Figure 5. Why is it worth showing that mouse and human ferrodoxins have similar activities/properties?

Author Response

Please see the attached file for the response to Reviewer #1.

Reviewer 2 Report

Comments and Suggestions for Authors I believe it has a positive impact on a specific field by studying the importance of isoforms in cell survival and cycle. I have no suggestions regarding the methodology used, but perhaps evaluating the interaction or impact on cyclins using Western blotting would be a good option.

Why is the comparison with other mitochondrial diseases not considered relevant? Are these isoforms feasible to study in another cell model? If so, which one?

Author Response

Please see the attached file for the response to Reviewer #2.

Reviewer 3 Report

Comments and Suggestions for Authors

The manuscript of “Dissecting the Biological Functions of Various Isoforms of Ferredoxin Reductase for Cell Survival and DNA Damage Response” examined different isoforms of Ferredoxin Reductase (FDXR), and the authors found the isoforms 1, 4 and 7 are the three most abundant forms in different types of cell. They showed that isoform 7 is localized in the cytosol, while 1 and 4 are majorly localized in mitochondria. The authors also generated several different cell lines to investigate each isoform’s function. Their results suggested each isoform of FDXR contributes to cell growth and the response to DNA damage, while the isoform 7 is sufficient for cell survival which might be due to its (unclear yet) extra-mitochondrial functions. The manuscript is well written, and data is fully supporting their conclusions. 

1, line 51, the font of FDXR need be changed,

2, line 61, change “Crispr-Cas9” to “CRISPR-Cas9”,

3, in Fig.1D, make the Y-axis be consistent as in Fig.1C and 1E, the current version is a little bit mis-leading.

4, in Fig.1H, the Mito Tracker and Flag labels need be switched,

5, the authors clearly made an abbreviation for “Mitochondrial localization signal (MLS)” in the lines 211-212, there is no need to repeat it in the following text, e.g. in the lines 246-247, line 250.

6, in Fig.2A, CPT treatment data need be added to keep the whole set of data to be consistent,

7, Fig.4D, remove the label “40” from the picture. Double check the labeling in the figure, make “Scr” or “scr” be consistent in all the panels. 

Author Response

Please see the attached file for the response to Reviewer #3.

Reviewer 4 Report

Comments and Suggestions for Authors

The manuscript entitled "Dissecting the Biological Functions of Various Isoforms of Ferredoxin Reductase for Cell Survival and DNA Damage Response seeks to investigate the association of the FDXR isoforms with the different cellular functions such as cell viability/survival and DNA Damage response. The manuscript is really interesting, well-organized and -written. However, I have two concerns (one minor, one major). 

Minor:

Authors need to provide within the manuscript the aim of this study, demonstrating the hypothesis-driven approach. 

Major:

I strongly recommend to perform Spearman's rho analysis by using publicly available data of different types of cancer from TCGA. First, you could create a list of the different molecules implicated in various signaling pathways (oxidative stress, cell survival, DDR) and then perform Spearman's rho using TCGA data (RNA-seq and mass spec or array data for breast cancer, hepatocellular carcinoma, colon cancer) to evaluate the potential association (positive or negative) of FDXR with the different pathway-related molecules. If possible, you could perform Spearman's rho for each FDXR isoform. Otherwise, you could use the total  FDXR expression. 

Author Response

Please see the attached file for the Response to Reviewer #4.

Round 2

Reviewer 1 Report

Comments and Suggestions for Authors

Thank you for addressing my comments.

Reviewer 4 Report

Comments and Suggestions for Authors

Dear Authors! Thank you very much for taking into account my comments. I have read the supplementary materials and found them really helpful.